# The Characteristics of Mercury Flux at the Interfaces between Two Typical Plants and the Air in *Leymus chinensis* Grasslands

**DOI:** 10.3390/ijerph181910115

**Published:** 2021-09-26

**Authors:** Zhaojun Wang, Xu Li, Gang Zhang, Lei Wang, Weihao Qi, Zhili Feng, Tingting Xiao, Mengping Yan, Deli Wang

**Affiliations:** 1School of Environment, Northeast Normal University, Changchun 130117, China; wangzj217@nenu.edu.cn (Z.W.); lix896@nenu.edu.cn (X.L.); wangl788@nenu.edu.cn (L.W.); qiwh669@nenu.edu.cn (W.Q.); xiaott070@nenu.edu.cn (T.X.); yanmp195@nenu.edu.cn (M.Y.); 2State Environmental Protection Key Laboratory of Wetland Ecology and Vegetation Restoration, Changchun 130117, China; 3Key Laboratory of Vegetation Ecology, Ministry of Education, Northeast Normal University, Changchun 130117, China; z35976113@163.com; 4Institute of Grassland Science, Northeast Normal University, Changchun 130022, China; fengzl306@nenu.edu.cn

**Keywords:** mercury, flux, *Leymus chinensis*, *Setaria viridis*, vegetation, Songnen Grasslands

## Abstract

Mercury is a global pollutant. The mercury exchanges between vegetation and the atmosphere are important for the global mercury cycle. Grassland ecosystems occupy more than 25% of the global land area and have different succession processes and ecological functions. The current research regarding mercury exchanges between forests and the atmosphere have attracted much attention, but the research regarding grasslands tends to be rare. To reveal the characteristics of mercury exchanges in grasslands, this study conducted field in-situ monitoring experiments in a *Leymus* meadow grassland regions of the Songnen Plains in northeastern China. The exchange flux values of the GEM (gaseous element mercury) between the plants and the atmosphere were measured using a dynamic flux bag method (DFB). The experiments were conducted for the purpose of assessing the mercury flux levels between the vegetation and the atmosphere in a typical *Leymus chinensis* meadow. The goal was to further the understanding of the change characteristics and influential factors and to describe the source and sink actions and dynamics between the grassland vegetation and the atmosphere. The diurnal variation characteristics were as follows: High during the day and low at night, with peaks generally appearing at noon. The growing period was characterized by absorption peaks of atmospheric mercury by the plants. The breeding period was characterized by the peak release of atmospheric mercury by the plants. The change characteristics were as follows: During the growing period, the duration of the plants in a mercury absorption state exceeded 96.5%, which was represented as the net sink of the atmospheric mercury. During the breeding period, the time of mercury release ranged between 46.4% and 66.8%, making the breeding period the net source of atmospheric mercury. The results of this study’s analysis indicated that each environmental factor was correlated with the mercury flux, and the environmental factors had different effects on the mercury flux during the different stages of plant growth. The atmospheric mercury concentration levels were the main factor during the growing period. Atmospheric humidity was the main factor during the breeding period. Solar radiation was the decisive factor during the entire experimental period.

## 1. Introduction

Mercury is considered to be a global pollutant [1]. Mercury entering the atmosphere can remain for between 0.5 and 2 a [2]. It may also accumulate to form atmospheric mercury reservoirs, be transmitted over long-distances through the atmosphere, and then sink in remote areas [3,4]. Mercury has become one of the most toxic heavy metals in the Earth’s environment [5]. The Minamata disease incident in the 1950s revealed mercury accumulation along the food chain and the process and mechanism of its toxic effects on humans and animals [6], thereby opening a prelude to today’s research on the health risks of mercury pollution. At the present time, the academic community is committed to establishing a global mercury cycle mass balance model [7]. Although the biogeochemical processes of mercury and methylmercury in aquatic ecosystems are now clearly understood, the research regarding the ecological environmental behavior of mercury in terrestrial ecosystems is currently in still advancing [8].

During the past several decades, academic circles have continuously deepened their understanding of the sources and sinks of atmospheric mercury [7,9]. For example, the atmospheric mercury consumption phenomena observed in the polar and sub-polar regions have indicated that gaseous element mercury (GEM) can be easily removed from the atmosphere through the oxidation induced by reactive halogens, resulting in 300 Mg·yr^−1^ atmospheric mercury sinks in the Arctic region [10,11]. Meanwhile, similar phenomena have been found to occur to lesser extents at mid-latitude oceanic interfaces [12]. The oxidation of GEM by ozone, active halogens, and hydroxyl radicals has also been observed in the free troposphere [13,14,15]. These findings have indicated that the residence time of GEM in the atmosphere may be shortened under certain environmental conditions [7], thereby sinking from the atmosphere to the land and sea. The dry subsidence rate (Vd) of GEM is mainly affected by such factors as surface characteristics, meteorological factors, soil, and water biochemical conditions. However, the Vd of bare soil and water surfaces is usually very small (less than 0.03 cm·s^−1^) and lower than the emission and re-emission of GEM from those surfaces [16]. Therefore, soil and water bodies are considered to be the net sources of atmospheric mercury [17]. In contrast, a large amount of dry deposition of GEM often occurs on vegetation-covered surfaces and wetlands and is often as high as 2 cm·s^−1^ [16]. Therefore, it has also been indicated that areas covered by vegetation may be important net sinks of atmospheric mercury.

Obrist [18] previously proposed that plants play a role in enriching atmospheric mercury and estimated that the annual absorption of atmospheric mercury global by vegetation may be as high as approximately 1024.2 t. Among the various vegetation types, the absorption of forest vegetation is estimated at 376.2 t, and the absorption of grassland vegetation is estimated at 648 t. Wang et al. [19] also estimated that the current global terrestrial ecosystem is a net mercury sink of approximately 8.5 Mg·yr^−1^. In recent years, seasonal field studies on GEM fluxes in forests, grasslands, and the tundra have also shown that those terrestrial ecosystems form a large number of annual net sinks of GEM in the atmosphere (2 to 20 μg·m^−2^·yr^−1^) [20,21,22]. At the present time, it is widely believed that in the mercury input of vegetation ecosystems, the absorption of atmospheric mercury by plant leaves is more dominant than that of dry and wet deposition [23,24]. GEM can enter plant leaves through stomata. However, due to its mild lipophilic properties, mercury may also enter plant leaves through the epidermis under certain conditions [25]. Many factors can affect the absorption processes of atmospheric mercury by plants. For example, as the plant tissue ages, the absorption through the epidermis decreases. Therefore, due to the nature of epidermal changes, the epidermal absorption sites become saturated over time [26]. Similarly, the exchanges of plant bodies at the epidermal level are also more sensitive to environmental factors. As a result, temperature, moisture, and light conditions can all potentially affect the mercury exchange fluxes between plants and the atmosphere by affecting stomatal conductance [27,28]. For example, it has been observed that during the years with low precipitation and high temperature levels, the concentration levels of mercury in the litter of the broad-leaved and coniferous forests of Huntington Forest tended to be low. This may be caused by the closure of stomata and the decreases in water vapor pressure [29]. In the month of June, the net precipitation in the Huntington Forest and the total mercury flux in the soil runoff were both at high levels [30]. However, in the artificially warmed and humidified permafrost soil on the Qinghai-Tibet Plateau of China [31], as well as that of Michigan and Minnesota in the United States, high Hg^0^ release levels have been observed. Furthermore, similar observations have been made in artificially warmed peatland soil [32], forested soil with canopies opened after logging activities [33,34], and forested areas following wildfires [35].

Although the global vegetation as a whole acts a net sink of atmospheric mercury, recent studies have found that under certain conditions, vegetation may also be a source of atmospheric mercury. For example, the examination of two subtropical coniferous forests in southern China [36,37] and a mature hardwood forest in Tennessee, United States [38], based on their annual flux calculations, showed that they were also net sources of GEM. Similarly, tree canopies and grasslands undergoing senescence or drought stress conditions have been observed to intermittently behave as GEM sources [39,40]. In the New England region (US), various types of reed vegetation that thrive in the freshwater environments of the coastlines were found to emit large amounts of mercury in their sediment during the summer seasons [41]. The aquatic vascular plants of the Florida Everglades have been confirmed to both absorb and release large amounts of atmospheric mercury. In addition, part of the release pulse has been found to be consistent with CH_4_, which may originate from the fibrous rhizosphere pool [42].

Recent studies regarding mercury fluxes of cattails and sawgrass [43,44] found that the daily release fluxes of those two types of plants were an order of magnitude higher than those of the lower water surfaces. The mercury flux measurements of marsh plants have revealed that in moderate mercury polluted wetlands [45] and primitive wetlands [46], the daily GEM flux was bidirectional. In other words, characterized by both deposition and release.

However, in the current research regarding the balance between atmospheric mercury reservoirs and the global mercury biogeochemical cycle, less attention has been paid to the mercury exchange processes between the surface vegetation and the atmosphere. The relevant research was observed to mainly focus on forest ecosystems, with the aspects of grassland vegetation and atmospheric mercury fluxes investigated in only a few reports to date. However, due to the differences in mercury absorption sites, as well as the different life histories and growth rhythms between grassland plants and forest plants, there are also obvious differences in the characteristics of mercury exchange fluxes between the two ecosystems and the atmosphere. Therefore, such factors should be studied and characterized separately. For example, the aforementioned two types of plants are characterized by different organs that absorb atmospheric mercury. Obrist [18] summarized the mercury pools of terrestrial vegetation and found that mercury concentrations could be detected in the leaves and branches of trees. Meanwhile, in regard to the mercury concentrations between grassland plants and the atmosphere, the exchanges almost entirely involved the leaves. In addition, the litter states of the two types of plants were different. For example, the leaves of tree canopies absorb atmospheric mercury during the growth period, and the litter formed becomes an important part of the atmospheric mercury dry deposition processes [47]. However, when grassland plants wither, the entire above-ground parts often form dead plant bodies and all return to the soil [48]. In addition, in terms of human factors, forest ecosystems are mainly affected by logging activities [49], while the grassland ecosystems tend to be mainly affected by grazing. Therefore, the impacts of human mowing and farming activities may have different influential effects. In order to clarify the process of mercury cycling in grassland ecosystems, in-depth research will be required from the following aspects: 1. Attention should be paid to the effects of grassland vegetation as a sink of atmospheric mercury on the reductions in regional atmospheric mercury concentration levels and 2. The effects of changes in the types of land usage and community succession on mercury fluxes in grassland vegetation and the atmosphere.

Grasslands are one of the main types of ecosystems in the world. Broadly speaking, grasslands include all types of herbaceous vegetation. The four main types of grassland belts in the world are the Steppe grasslands, Prairie grasslands, Pampas grasslands, Savana grasslands. According to the current definition, the global grassland area measures 5.25 × 10^7^ km^2^, which accounts for 40.5% of the total land area on Earth [50]. The experimental focus of this study was located in the center of China’s Songnen Grassland region. The Songnen Grassland region is situated on the eastern edge of the steppe vegetation continuous belt of Eurasia. The region is characterized by a mid-temperate continental semi-humid and semi-arid climate, with an area of 1.7 × 10^5^ km^2^. The vegetation type is mainly *Leymus chinensis* (Dian) [5]. This study’s experimental subjects included *Leymus chinensis* (Trinius ex Bunge) Tzvelev and *Setaria viridis* (Linnaeus) P. Beauvois. It has been determined that *L. chinensis* is a constructive species in the study area. It was found to be widely distributed in the study area and had a strong representativeness. In addition, *Setaria chinensis* is also a common species in the study area and was previously used as a typical experimental plant in the existing research regarding carbon and nitrogen fluxes of the aforementioned ecosystem. Therefore, due to its widely reported findings [51], this study considered that it would be an appropriate representative of the weed varieties in the focus area. This study collected data regarding the plant and atmospheric mercury fluxes and related environmental factors in the field. The goals of this study were as follows: (1) To determine the atmospheric mercury concentration levels in the *L. chinensis* meadow area of the Songnen Grassland region; (2) To analyze and understand the typical and common plants and atmospheric conditions of the *L. chinensis* grasslands, including the characteristics of the day and night changes in the mercury exchange flux and clarification of its influential factors; and (3) To discuss the dynamics of vegetation as a mercury reservoir between the sources and sinks during different growth periods and atmospheric conditions.

## 2. Materials and Methods

### 2.1. Overview of the Study Area

This study’s experimental site was located in Changling County of western Jilin Province, China (Figure 1), in which the Northeast Normal University Songnen Grassland Ecological Research Station is situated (44°45′ N, 123°45′ E). The regional climate type is a semi-arid temperate continental climate, with average annual rainfall levels lower than the evaporation levels. Due to long-term human disturbances (for example, overgrazing and reclamation) and changes in natural factors, the grasslands of the research site were in the process of community retrograde succession, which was characterized by soil bulk density and pH increases and moisture and organic matter decreases. The dominant population of *L. chinensis* was observed to exhibit a decreasing trend. Therefore, the dominance of the dominant population of *L. chinensis* was essentially in decline. However, the relative coverage of the halophyte population was observed to be significantly increased. Three types of communities with different degrees of degradation were formed in the region as follows: Pure *L. chinensis* communities; *L. chinensis* plus weeds communities; and weed grass communities.

The region in which the study area was located has the characteristics of a semi-arid temperate continental climate. The winter seasons are cold and long, with little snowfall, summer seasons are warm and rainy, and springtime features windy and dry conditions. The extreme minimum temperature is −40.3 °C, and the extreme maximum temperature is 38.9 °C. The annual average temperatures range between 1.5 and 4.2 °C. The sunshine duration averages approximately 2880 h. The annual average windy days generally exceed 100 days, and the annual average wind speed is approximately 5.7 m/s. The average annual rainfall is 430 mm, with the majority of the rainfall concentrated from June to September. The precipitation during those months can reach 60 to 80% of the annual rainfall. The average annual evaporation is approximately 1600 mm, which is three to four times that of the rainfall.

The terrain in the study area was observed to be flat and low, and the soil structure was intricately inlaid. This study found that the region was dominated by chernozem, alkaline, meadow, and aeolian sandy type soil. The remainder was determined to be saline and swamp soil. The groundwater resources were observed to be highly mineralized. Since the terrain was low, the drainage was not smooth. Therefore, it was considered that the area was semi-arid, with the evaporation rate greater than the precipitation. Due to the aforementioned factors, the salt could not be smoothly discharged and collected in the soil. At the same time, the long-term agricultural grazing behaviors have affected the soil structures of the vegetation in the area, resulting in serious destruction. Therefore, the soil salinization in the Songnen Grassland area is becoming increasingly more serious and the affected areas are expanding. The pH values of the soil in the study area in this research investigation were determined to range between 7.5 and 10.5.

The type of grassland examined in this study was meadow grassland. The area of that type of grassland measured 1.2 × 10^6^ hm^2^, which is 55.8% of the total area of the Songnen Grassland region. Meadow grass is the main grassland type of Songnen Grasslands. Among the types of meadow grass, *L. chinensis* + forb has been determined to be the largest community type, reaching 8.4 × 10^5^ hm^2^, accounting for 70.3% of this type of grassland. The plant compositions of meadow grasslands are mainly xerophytic perennial rhizome grasses and clump grasses, among which there are a variety of plant species. The dominant and most common species include *L. chinensis* (Trinius ex Bunge) Tzvelev; *Filifolium sibiricum* (Linnaeus) Kitamura; *Chloris virgata* Swartz; *Setaria viridis* (Linnaeus) P. Beauvois, and so on.

Among the aforementioned species, *L. chinensis* and *S. viridis* were the experimental subjects selected for the current investigation.

*Leymus* of the *Gramineae* family is a perennial C3-positive plant with high vegetative reproduction and low seed yield. It is characterized by a fast growth rate; high drought resistance; cold resistance; and alkali resistance. The plants of this species generally grow between 40 and 90 cm in height, with 4 to 5 internodes. The leaves range between 7 and 18 cm in length and 3 to 6 mm in width. The leaves are flat or involute, rough on the top and edges, and smooth on the bottom. On the Songnen Plain, this variety begins to sprout and turn green in early April; heading occurs in late May; flowering in June; and fruiting in July and August. As the top community, its coverage on the Songnen Grassland region may reach 65%. *L. chinensis* reproduces mainly by asexual reproduction that relies on adventitious buds. In the case of asexual reproduction, it has been found that plants tend to more sensitive to environmental factors.

*Setaria* of the *Gramineae* family is characterized by a high protein content. It has been found to adapt well to complex environmental conditions, with high resistance to cold, heat, pests, and diseases. The *Stetaria* variety tends to grow quickly and displays strong reproductive abilities. The culms are upright or geniculate at the base, ranging between 10 and 100 cm in height. It has been observed to have flat blades and long acuminate or acuminate apexes with obtusely rounded bases, which are almost truncated or narrow (4 to 30 cm long; 2 to 18 mm wide). The plants are usually glabrous or covered with sparse wart-like hairs with rough edges. The flowering and fruit periods range from May to October. Generally speaking, *Stetaria* has low requirements for soil conditions and can easily compete with other species.

### 2.2. Research Methods

#### 2.2.1. Sample Plot Layout

The experimental sample plot in the study area was located in the Beidianzi Grasslands (Figure 2), which is a low-lying area of the Songnen Grassland region. The study area was determined to be composed of flat land, swamp areas, water surfaces, and sand dunes. It was observed to have a microwave-like flat terrain. A flat and vast meadow grassland was located between the dunes. It was determined that the vegetation type had the characteristics of a meadow grassland, and the vegetation division belonged to the category of a forest grassland area. In addition, the agriculture belonged to that of an interlaced zone of agriculture and animal husbandry. The zonal vegetation was found to be mainly the *L. chinensis* community.

This study’s sample point layout is detailed in Figure 2. A 50 m × 50 m sample plot was randomly selected in the study area. The sample plot was divided into one hundred 5 m × 5 m plots using a grid method. Then, a random vegetation survey was carried out in the sample plots. The surveyed sample plots were first divided into three groups according to the community type as follows: *L. chinensis* community (*L. chinensis* coverage > 80%); *L. chinensis* (40% < *L. chinensis* coverage < 60%) + weeds (40% < miscellaneous grass coverage < 60%) community; weeds community (weed coverage > 80%). In the *L. chinensis* + weed community, one of the samples was selected for this study’s subsequent experiments using a simple random sampling method. In addition, one of the *L. chinensis* and *S. sylvestris* plants in the sample was randomly selected for this study’s in-situ experiments.

The field experiments and sampling completed in 2019 were divided into two periods: as follows: 1. Vegetative reproduction period (June and July); and 2. Reproduction period (August) [52,53]. A total of six data collection processes were carried out. The plant samples were subjected to three measurements of mercury flux under different growth period conditions. Each measurement duration was a continuous 24-h period, and the required field positioning experimental time was 144 h. Previous research has indicated that foliar mercury (Hg) flux is bi-directional, with influence from both atmospheric and soil Hg.

#### 2.2.2. Sampling and Measurement Methods

##### Sampling Method

Within the study plots, the selected soil was covered by a flux bag under the plant. A drill was used to sample 0 to 2.5 cm of surface soil. The samples were immediately bagged and labeled and then transported to this study’s laboratory facilities. The samples were air dried for seven days and then filtered using an 80-mesh nylon sieve. Following the in-situ experiments, the above-ground parts of the samples were harvested and taken back to the laboratory for weighing (dry weight) and the measurements of plant height. The Blackman Formula was used to calculate the relative growth rates of the two plants. The formula was as follows:(1)$$\mathrm{RGB}\;=\;\frac{\ln(W_{i+1})\;-\;\ln(W_{i})}{T_{i+1}\;-\;T_{i}}$$

In the formula, W_i_ represents the dry weight at T_i_; W_i+1_ indicates the dry weight at T_i+1_; and the formula represents the relative growth rates of the plants from T_i_ to T_i+1_.

##### Testing Methods

In the current study, a Lumex RA-915^+^ mercury analyzer (originating from Russia; detection limit of air samples: 2 ng/m^3^) combined with a flux bag method was used to determine the mercury exchange fluxes between the plants of the study area and the surrounding air. A Tedlar dynamic flux bag (length × width = 600 mm × 450 mm; minimum volume 20 L) was selected as the flux bag. Tedlar dynamic flux bags are often used in atmospheric mercury exchange research due to their high durability, flexibility, and radiant transparency in the photosynthesis spectrum [45,54,55]. The aforementioned mercury analyzer was equipped with a built-in air pump. During the sampling processes, the air entered the flux bags via air inlets and then flowed out through air outlets under the action of an air pump in order to form a gas flow. The gas flow rate was determined as the minimum flow rate when the mercury concentration differences between the inlet and outlet were stable [56]. The differences between the mercury concentrations at the inlets and outlets of the empty bags were measured and compared in order to determine the blank.

During the measurement process, the flux bags were used to cover the outsides of the plant bodies and were sealed at the lowest end. The air inlets of the flux bags were connected to the air outlet of the RA-915^+^ mercury analyzer. Monitoring was conducted once every 10 s, with 30 data points recorded every 5 min. The average values were then calculated and recorded as C_in_. The unit was ng/m^3^. After 5 min, the flux was changed to the outlets of the bags, which were connected to the inlet of the RA-915^+^ mercury analyzer. The gaseous mercury concentrations in the outlet gas were monitored every 10 s, with 30 data points recorded every 5 min. The average value was calculated and recorded as C_out_ in ng/m^3^. Alternating data measurements were continuously collected over a 24-h period. A modified standard flux box equation was used for the flux bags in order to calculate the leaf surface exchange flux. The calculation formula was as follows:F = (C_out_ − C_in_) × Q ÷ A(2)

In the formula, F represents the calculated mercury exchange flux per unit of leaf surface from the atmosphere to the plant (ng/(m^2^·h); C_in_ is the gaseous mercury concentration in the air at the inlet of the flux bag (ng/m^3^); C_out_ indicates the concentration of gaseous mercury in the air at the outlet of the flux bag (ng/m^3^); Q denotes the gas flow rate through the flux bag (m^3^/h); and A is the total leaf area of plants in the flux bag (m^2^).

In order to reduce the chance of error, the differences between the average value of two consecutive C_out_ measurements and the average value of four C_in_ (before and after) were taken. Therefore, if the calculated flux value was positive, the mercury exchange process indicated that the plants had released mercury into the atmosphere. However, if the flux value was negative, the mercury exchange process indicated that the plants had absorbed mercury from the atmosphere.

This study utilized a LUMEX RA-915^+^ coupled with a UMA (solid–liquid mercury analysis unit; detection limit: 1 ng/kg) mercury analyzer for the purpose of determining the concentration levels of soil mercury. The detection range was between 0.5 mkg/kg and 0.5 mg/kg. The measurement time for one sample was between 50 and 70 s. The sample sizes ranged from 10 to 300 mg, and the standard deviation of the baseline signal measurement value was 2 ng/m^3^. Each soil sample was subjected to three parallel tests, and the data were recorded.

A leaf area meter was used to measure the leaf areas of the samples in the field. In addition, a portable weather monitor (ZX-SCQ4, Beijing, China) was used to record the hourly solar radiation, air temperature, air humidity, soil temperature, humidity, and other data.

#### 2.2.3. Data Analysis Method

SPSS Statistics 23 (Armonk, NY, USA) was used for the data analysis, and Origin Pro 8 (OriginLab, Guangzhou, China) was used for the graphing processes in this study. The Pearson correlation coefficient was used to statistically test the correlations between relevant environmental factors and the plant/atmospheric mercury fluxes. In addition, a path analysis method [57] was used to examine the importance levels of the various environmental factors.

## 3. Results

### 3.1. Soil Surface and Atmospheric Mercury Content Levels

This study’s field in-situ experiments involving the *L. chinensis* and *S. sylvestris* samples were carried out on two adjacent days (48-h period) each month. Table 1 shows that the near-ground atmospheric mercury concentration levels measured with *L. chinensis* as the sample experiment ranged among 6, 7, and 8. The daily average value and range in August were 16.6 ± 6.9 ng·m^−3^ (4.0 to 28.0 ng·m^−3^); 16.6 ± 6.6 ng·m^−3^ (7.0 to 28.0 ng·m^−3^); and 16.7 ± 6.5 ng·m^−3^ (8.0 to 28.0 ng·m^−3^), (n = 289), respectively. The daily average values and ranges of the mercury concentrations in the near-surface atmosphere measured in the months of June, July, and August when using *Setaria* as the sample experiment were 16.4 ± 6.0 ng·m^−3^ (7.0 to 26.0 ng·m^−3^); 16.4 ± 6.0 ng·m^−3^ (7.0 to 26.0 ng·m^−3^); and 13.8 ± 4.2 ng·m^−3^ (8.0 to 21.0 ng·m^−3^), (n = 289), respectively. During the experiments, the near-surface atmospheric mercury concentration levels remained stable, were significantly higher than the global background value (1.5 to 2.0 ng·m^−3^) [58], the relevant reported value of the Neustift temperate plain grassland ecosystem in Central Europe (1.7 ± 0.5 ng·m^−3^) [59], and the reported value of an Australian alpine grassland (0.54 to 0.63 ng·m^−3^) [39]. The levels were also higher than those of the alpine forest of the Changbai Mountain area at the same latitude (3.22 ± 1.78 ng·m^−3^).

As detailed in Table 2, the average surface soil mercury concentration levels in the sample plots during the months of June, July, and August were 16.3 ± 1.5 ng·g^−1^, 6.6 ± 0.7 ng·g^−1^, and 8.9 ± 1.0 ng·g^−1^, respectively. Those values were determined to be lower than the background value of the soil mercury in the Songnen Grasslands (19.56 ng·g^−1^). However, when compared with the relevant reported value (14.9 ± 10.4 ng·g^−1^) in the Steppe Grassland region of Inner Mongolia, the concentration levels were determined to be higher in June, but lower in July and August.

### 3.2. Mercury Exchange Flux at the Plant/Atmosphere Interface

The mercury exchange flux levels of the two types of plants were observed to be basically the same between June and July, as shown in Table 3. The main mercury exchange processes of the two types of plants were all manifested as the plants absorbing mercury from the atmosphere. In the 24-h measurement results, it was determined that the time of mercury absorption by *L. chinensis* exceeded 99.6%, the time in which *S. viridis* absorbed mercury was greater than 96.5%, and the mercury emitted from plants into the atmosphere was minimal (n ≤ 10). The net absorption levels of the two types of plants in June were slightly higher than those observed in July. The net values and ranges of the daily average exchange fluxes were as follows: *L. chinensis* in June: −1.58 ng·m^−2^·h^−1^ (−5.76 to 0.52 ng·m^−2^·h^−1^); *Setaria* in June: −1.54 ng·m^−2^·h^−1^ (−5.42 to 1.67 ng·m^−2^·h^−1^); *L. chinensis* is in July: −1.19 ng·m^−2^·h^−1^ (−4.97 to 0 ng·m^−2^·h^−1^); and *Setaria* in July: −1.37 ng·m^−2^·h^−1^ (−5.31 to 1.01 ng·m^−2^·h^−1^).

In the month of August, both species displayed a net release process from the plants to the atmosphere. The release rates of *S. vulgaris* were greater than those of *L. chinensis*. The net values and ranges of the daily average exchange fluxes were as follows: *L. chinensis*: 0.34 ng·m^−2^·h^−1^ (−1.94 to 4.84 ng·m^−2^·h^−1^) and *Setaria*: 1.18 ng·m^−2^·h^−1^ (−0.91 to 4.55 ng·m^−2^·h^−1^). In addition, in the 24-h measurement results, it was observed that the time in which *L. chinensis* released mercury accounted for 46.4%, and the time in which *S. vulgaris* released mercury accounted for 66.8%.

### 3.3. Plant Indicators

The measured biomass, plant height, and leaf area of the experimental plant samples, along with the recorded monthly fluxes, were used as the plant indicators in this study, as detailed in Table 4. The three indicators of the two types of plants were observed to increase significantly during the months of June and July. There were no increases in the biomass and plant heights during the month of August. However, the leaf areas increased slightly during that period. Therefore, it was determined that the two examined plant types in the study area were in a vegetative reproductive stage during the months of June and July and then entered a reproductive stage in August.

The RGB calculation results are shown in Table 5. It can be seen in the table that the relative growth rate of *L. chinensis* from June to July was 0.0055, and the relative growth rate from July to August was −0.0005. The relative growth rate of *Setaria* from June to July was 0.0051, and the relative growth rate from July to August was 0.0051. Overall, the relative growth was approximately −0.0012. Therefore, it was indicated that the two examined plant types experienced obvious growth between June and July, and the growth become less obvious or stopped in August.

## 4. Discussion

### 4.1. Variation Characteristics of the Plant and Atmospheric Mercury Fluxes

#### 4.1.1. Changes in the *L. chinensis* Plant and Atmospheric Mercury Fluxes

Two-way mercury exchange fluxes were observed during the study period. The daily average value was −0.81 ± 1.55 ng·m^−2^·h^−1^, and the mercury flux range was between −5.76 ng·m^−2^·h^−1^ and 4.84 ng·m^−2^·h^−1^. This study found that the flux values during the day were higher than those at night, and there were major differences between the two time periods. The observed mercury flux exchange processes between the plants and the atmosphere are detailed in Figure 3. It was found that during the vegetative reproduction phase (June and July), the valley value (maximum value of atmospheric mercury absorption by the plants) appeared at noon (12:00). From 8:00 to 9:00. the mercury fluxes began to decrease significantly. Then, the lowest values were reached between 12:00 and 13:00 before rising again. The rising of the mercury fluxes stopped between 16:00 and 17:00, at which point they were observed to fluctuate at approximately 0 to −2.62 ng·m^−2^·h^−1^. The reproductive stage (August) followed the diurnal variations of the peak (maximum value of the mercury released by the plants into the atmosphere) at noon (12:00), and then the mercury flux began to rise significantly between 6:00 and 7:00 in the morning. At 13:00, the peak was reached, after which the flux began to drop, stopped after 18:00, and then oscillated between 0 and −1.94 ng·m^−2^·h^−1^. It was worth noting that during the month of August, this was observed after 6:00 in the morning and 19:00 in the evening. At the point of mercury compensation, there were large differences observed in the atmospheric mercury concentrations at those two time points, as well as major differences in the atmospheric temperature and humidity. These findings indicated that multiple environmental factors had obvious comprehensive effects on the mercury flux during the reproductive stage. During this study’s experimental period, the mercury fluxes in the vegetative reproductive stages of *L. chinensis* were determined to be net deposition, and only one mercury flux release moment was observed during the flux monitoring for a total of 48 h in June and July. Meanwhile, during the reproductive stage of *L. chinensis*, it was determined that the mercury fluxes in August were net releases at low concentration levels since the release values during the day were slightly higher than the deposition values at night.

The flux levels of this study were found to be consistent with the values determined in previous related studies. For example, Fritsche et al. [60] observed a value of −4.3 ng·m^−2^·h^−1^ (−27 to 14 ng·m^−2^·h^−1^), and the observed value of the grassland on the Neustift Plain in Austria was −2.1 ng·m^−2^·h^−1^ (−41 to 26 ng·m^−2^·h^−1^). In addition, Converse et al. [61] observed a value of 2.5 ng·m^−2^·h^−1^ (−124.8 to 82.4 ng·m^−2^·h^−1^) in the Big Meadows prairie region (United States) during the summer, and Howard et al. [39] observed a grassland value of 0.2 ng·m^−2^·h^−1^ (−52.9 to 54.7 ng·m^−2^·h^−1^) in the Australian high mountains in the Southern Hemisphere. Therefore, it could be seen that in terms of the net exchange values of mercury flux, this study’s findings were basically at the same level as those of related grassland studies, while the variation ranges of the mercury flux were smaller than those of other similar studies.

The observed phenomena of large amounts of mercury deposits during the vegetative reproduction stage of the examined plants during the months of June and July were basically consistent with the conclusions reached in other related studies, such as the seasonal studies conducted by Fritsche et al. [59] and Converse et al. [61], and the modeling studies conducted by Hartman et al. [62]. Generally speaking, researchers in the field all believe that during the growing season of plants, the deposition of GEM on vegetation will increase. Howard et al. [39] proposed that mercury deposition will also increase with the increases in plant biological activities throughout the year. The slight difference was that in this study, there was almost no mercury flux released by the examined plants to the atmosphere during the plant vegetative reproduction stage (June and July). However, the results of many related studies [39,60,61] indicated that the diurnal variations in mercury flux were composed of a net release during the day and net deposition at night. This was consistent with the variations in the mercury fluxes during the month of August in this study. The differences in this study’s results may have been caused by a number of factors. For example, the atmospheric mercury concentration levels reported by Howard et al. [39] ranged between 0.54 and 0.63 ng·m^−3^, and the atmospheric mercury concentration levels reported by Fritsche et al. [54] ranged from 1.2 to 1.7 ng·m^−3^, which were significantly lower than the atmospheric mercury concentration measured in this study. Therefore, a higher concentration of atmospheric mercury (Table 1) may have caused large amounts of mercury to be deposited from the atmosphere on the plants [63]. Howard et al. [39] pointed out in their study that the grassland plants in the study area were in a state of senescence. However, the plants used in this study were in a rapid vegetative reproduction stage in June and July, with higher biological activities, and at the same time, under relatively high atmospheric conditions. High mercury concentration levels may cause plants to absorb atmospheric mercury more strongly. Although the study conducted by Fritsche et al. [60] did not mention whether the examined plants entered the breeding period, the daily variations in atmospheric mercury concentration levels were reported to be higher in the morning and evening, reaching the lowest value at noon. This was contrary to the observations made in this study. It also showed that the atmospheric mercury concentrations may be a dominant factor influencing mercury flux during plant vegetative reproduction.

#### 4.1.2. Variation Characteristics of *S. sylvestris* Plants and Atmospheric Mercury Fluxes

During the study period, two-way mercury exchange fluxes between *Setaria* and the atmosphere were observed. An average value of −0.57 ± 1.83 ng·m^−2^·h^−1^ was determined, and the mercury flux ranged between −5.42 ng·m^−2^·h^−1^ and 4.55 ng·m^−2^·h^−1^. The flux values during the day were greater than those at night, and there were major differences observed between the two. The mercury flux exchange processes between the plants and the atmosphere are shown in Figure 4. During the vegetative reproduction phase (June and July), following the diurnal variations of the valley at noon (Figure 4), the mercury flux began to become significant after 9:00. A decline followed, in which the mercury flux was observed to drop to the lowest point from 12:00 to 13:00 (noon) and then increase again before stopping at approximately 17:30. At that point, the mercury flux was observed to oscillate between 0 and −2.08 ng·m^−2^·h^−1^. The vegetative and reproductive stage (August) followed the characteristics of the diurnal variations with peaks observed at noon. It was found that the mercury flux began to rise significantly at approximately 5:00 in the morning; reached a peak after 13:00; then decreased and stopped after 21:00. The mercury flux then oscillated between 0 and −0.91 ng·m^−2^·h^−1^. Similar to *L. chinensis*, mercury compensation points were observed in the morning and evening during the month of August, but they appeared earlier in the morning (approximately 5:00) and later at night (approximately 22:00). The atmospheric mercury concentration levels were also observed during these two periods. The differences indicated that during the reproductive stage of *Setaria*, there were also important influential factors other than the atmospheric mercury concentrations. This study found that throughout the research process, the mercury flux values during the vegetative reproduction stage of *S. viridis* were net deposition. It was determined that during the total 48-h flux monitoring period (n = 289) in June and July, only ten periods of mercury flux from *S. viridis* to the atmosphere were observed. In addition, during the reproductive stage of *Setaria*, the mercury flux values in August were slightly different from those of *L. chinensis*, and the mercury emission flux was at the same level as that in the absorption stage.

During the vegetative reproductive stage, the mercury flux changes of *Setaria* and *L. chinensis* were roughly the same. During the reproductive stage, both *L. chinensis* and *Setaria* showed a change pattern of net release during the day and net deposition at night. Many previous studies have considered that the aforementioned change pattern appeared to be related to changes in humidity levels [21,38,63], with the higher solar radiation and temperature levels during the day causing the humidity to decrease. However, during the night, as the humidity increases, atmospheric mercury is deposited on the surfaces of the leaves with moisture and then evaporates with the moisture after dawn to be released back into the atmosphere. That statement effectively explains the observed changing trends of *L. chinensis* and *S. chinensis* during the vegetative and reproductive stages of the present study. However, the mercury fluxes of *S. chinensis* and *L. chinensis* during the vegetative and reproductive stages were found to be slightly different, and the specific performance was that the net flux of *S. sylvestris* was higher than that of *L. chinensis*, which was determined to be caused by a longer mercury release time and higher levels of release flux in the morning and afternoon. Among the environmental factors (atmospheric humidity, atmospheric temperature, solar radiation, soil humidity, and soil temperature) monitored in this study, no significant differences were found between *L. chinensis* and *S. viridis* during the month of August. There were two plant stomatal density values in the studies regarding the water use efficiency and stomata density of *L. chinensis* and the leaf morphology of *S. chinensis* in the neighboring areas of the study area: *S. chinensis*: 935.41/mm^−2^ and *L. chinensis*: 100.5/mm^−2^. Therefore, it was considered that the differences in the stomata densities may have been the reason for the higher mercury release levels and longer mercury release times of *S. viridis* when compared with *L. chinensis*. At the same time, the stomatal densities of *L. chinensis* displayed a phenomenon of plant domestication dominated by water. The sample plot selected in this study was composed of saline–alkali soil, and there was no water stress observed. In that case, the differences in the mercury flux between the two plant types may have been the result of different stomatal densities.

This study compared the obtained results with those of other studies regarding grassland ecosystems [39,59,60,61] and found that the mercury fluxes observed in this study varied within a small range and that there was often an order of magnitude difference when compared with other related studies. However, in the mercury flux studies conducted by Marsik et al. [44] and Lee et al. [64], mercury flux variation ranges similar to those of this study were found. This may also have been caused by the saline–alkali stress conditions in the study area, which are known to often affect the water content and various physiological activities of plants, thereby affecting the deposition of atmospheric mercury on plants and the release of mercury from plants.

#### 4.1.3. Comparison of Mercury Flux Characteristics between the Two Plants and Atmosphere

In the current research investigation, the two types of plants did not show significant differences in the characteristics of their mercury flux changes. During the vegetative reproduction stage, both plant types showed net atmospheric mercury deposition within 24 h and very little plant mercury release. During the reproductive stage, both plant types displayed net release during the daytime and a pattern of net deposition at night. In terms of the mercury flux levels, the two plant types displayed subtle differences (Table 3). By separating the daily mercury deposition and release statistics, it was determined that the average daily mercury deposition and mercury release values of *S. viridis* were usually higher than those of *L. chinensis*. This may have been due to the fact that the mercury flux of *L. chinensis* tended to increase and decrease rapidly at approximately noon, while *S. chinensis* tended to maintain a more active mercury exchange state in the morning and afternoon, in addition to peaking at approximately noon (12:00).

The coverage of *L. chinensis* in the study area was considered to be relatively large at approximately 40 to 60%, thereby occupying a dominant position in the community. A variety of weeds represented by *Setaria* occupied the remaining space. It has been observed that mosaic communities will often form between *L. chinensis* and weed populations, resulting in clearly visible staggered population distribution areas in natural grasslands. Plant distribution patterns are affected by the interactions between plants and environmental factors. In the cases of high intensity levels of inter-species competition (middle stage of grassland degradation, or the current level), weed species tend to invade *L. chinensis* populations in agglomeration distribution patterns. In addition, the *L. chinensis* populations will resist invasions of other grass species in agglomeration distribution patterns [65]. At the present time, the overall plant density levels of the grasslands are relatively high, and their roles as mercury sinks are relatively strong. During the later stages of competition (late grassland degradation), the invasion of the *L. chinensis* population by weed populations is basically completed, and large-scaled coexistence can be observed. The subsequent interactions will lead to the transformation of the spatial distribution patterns of the plants from an agglomeration distribution to a random distribution pattern and then from a random distribution to a diffusion distribution pattern [66]. As a result, the overall plant density of the grassland will become lower, and its role as a mercury sink in the atmosphere will be reduced.

During this study’s experimental period, both *L. chinensis* and *S. chinensis* acted as net sinks of atmospheric mercury. However, due to the higher mercury release flux of *S. chinensis* during the reproductive stage, its role as a net sink was lower than that of *L. chinensis*. In the future, as the dominance of the *L. chinensis* population continues to decrease and the dominance of weeds (represented by *S. vulgaris*) continues to increase, as estimated from the changes in the population coverage, the overall role of grasslands as atmospheric sinks may decrease by 14.4 to 20.2%.

### 4.2. Correlation Analysis between the Mercury Flux and the Influential Factors

The influential factors identified as the controlling factors of the mercury exchange fluxes from the previous related research include humidity [67,68]; solar radiation [69,70]; temperature [70,71]; and atmospheric mercury concentrations [72]. In this study, the plant and atmospheric mercury fluxes and the measured environmental factors displayed significant linear correlations (Table 6), and the impact of each environmental factor on the two types of plants showed high consistency. The correlations were calculated by Pearson’s correlation coefficient, and the significance level was 1%. The level of the correlation coefficient was between −1 and 1 (1 means the variable is completely positively correlated, 0 means irrelevant, and −1 means completely negatively correlated).

Many previous studies have shown that the plant and atmospheric mercury exchange fluxes are affected by the atmospheric mercury concentration levels [73,74,75]. It is considered that the deposition of mercury on plants occurs at higher atmospheric mercury concentrations, as well as at lower atmospheric mercury concentrations when mercury is released from plants. In this study, at the plant vegetative reproduction stage, the atmospheric mercury concentration and mercury flux values were found to be significantly negatively correlated, which was manifested in the fact that as the atmospheric mercury concentration increased, the atmospheric mercury deposition on plants increased. These findings are in line with the conclusions obtained in other related studies. During the reproductive stage of plants, there is a significant positive correlation between the atmospheric mercury concentrations and the mercury flux values. The specific manifestation is that when the atmospheric mercury concentration increases, the mercury release flux from plants to the atmosphere also increases. That type of situation occurs due to the interactions of various environmental factors. For example, under the influential effects of solar radiation and temperature, the release of mercury in the soil will lead to increases in the concentration levels of mercury in the atmosphere near the ground. At the same time, the release of mercury on the surfaces of plants due to solar radiation and temperature will also have promoting effects, resulting in invisible impacts of atmospheric mercury concentrations on plant/atmospheric mercury fluxes. In addition, there is a possibility that the higher concentrations of mercury in plants will increase the mercury compensation point.

In mercury flux studies, solar radiation and temperature have been proven to each have their own independent effects. For example, solar radiation can cause the reduction of HgII, and increases in temperature can potentially volatilize the previously stored mercury. However, it has been observed in field experiments that due to the inherent connection between these two factors, it is difficult to resolve the mutual influences between them. In the current investigation, the effects of solar radiation and temperature (air temperature and soil temperature) on the mercury fluxes of plants and the atmosphere displayed the characteristics of consistency. For example, during the plant vegetative reproduction stage, it was significantly negatively correlated with the mercury flux. However, during the plant reproduction stage, it was found to have a significant negative correlation with the mercury flux. Therefore, the mercury flux was significantly positively correlated. The correlation between the three environmental factors and the mercury flux during the reproductive stage was stronger than that observed during the vegetative reproductive stage. At the same time, the results of the reproductive stage were also more consistent with the results of other studies. The increases in solar radiation and temperature resulted in increases in HgII on the surfaces of the plants. This resulted in reductions, and at the same time volatilization, of the mercury left by the previous deposits. Therefore, under those influential effects, the plants released mercury into the atmosphere. It was found that during the vegetative reproduction stage, with the increases in solar radiation and temperature, the deposition of the atmospheric mercury on the plant surfaces continued to strengthen. This may have been due to the large amounts of mercury released from the surface soil under the action of solar radiation and temperature, which caused higher levels of atmospheric mercury near the ground. Consequently, when the concentration levels rose, the photosynthesis, respiration, and other biological activities of the plants during the vegetative reproduction period were stronger, which caused plants to absorb more mercury from the atmosphere.

Humidity levels can affect mercury flux from multiple levels. For example, atmospheric mercury will be deposited on the surfaces of plants through moisture when humidity levels are high and evaporate from the surfaces of the plants under dry conditions, resulting in the differences between night dew and non-night dew [39]. The mercury in the soil will also rise to the surface along with the transpiration of water and then be released into the atmosphere [76]. This will lead to increases in atmospheric mercury concentrations and will affect the plant/atmospheric mercury flux values. Humidity is also an important factor affecting plant stomatal conductance [63]. In this study, the effects of soil moisture and atmospheric humidity on the two types of plants were found to be consistent, showing a significant positive correlation during the plant vegetative reproduction stage and a significant negative correlation during the reproduction stage. The negative correlation in the reproductive stage indicated that as the humidity increased, the release of mercury from the plants decreased. In addition, the increased deposition of atmospheric mercury on the plants was determined to be caused by the deposition of atmospheric mercury with water. In the vegetative reproduction stage, with the increases in humidity, it was found that the deposition of atmospheric mercury on the plants had decreased. This was caused by the negative correlations between the humidity levels and the solar radiation and temperature. The correlation coefficients are detailed in Table 6. As can be seen in the table, for the solar radiation, the correlation coefficients between the temperatures during the plant vegetative reproduction stage were higher than those of the humidity. This affected the correlation analysis results between the humidity and the mercury flux.

### 4.3. Path Analysis between the Mercury Flux and the Influential Factors

#### 4.3.1. Influencing Factors and Exchange Dynamics of the Mercury Flux in *L. chinensis*

As detailed in Table 7, this study’s path analysis results showed that during the vegetative reproduction stage of *L. chinensis* (June and July), the direct effects of various environmental factors in descending order were as follows: Atmospheric mercury concentrations; soil temperature; atmospheric humidity; solar radiation; and soil humidity. It was found that the addition of other variables did not significantly improve the regression model. Among the main factors, it was found that the direct influence of atmospheric mercury concentrations on the mercury flux was obviously dominant, which had strong negative effects (higher atmospheric mercury concentration led to higher mercury deposition flux). Among the other environmental factors, atmospheric humidity, solar radiation, and soil temperature were observed to have higher direct impacts. Meanwhile, the atmospheric humidity and solar radiation had negative effects, and the effects of soil temperature were positive.

The results of the path analysis also verified part of the predictions of the Pearson correlation analysis results and complementary to a certain extent. Higher atmospheric humidity often occurs at night when atmospheric mercury is deposited with water on plant bodies. Solar radiation indirectly affects the mercury concentration levels in the atmosphere and thereby the mercury flux. The direct influence coefficients of the solar radiation confirmed that it was a negative effect. Therefore, it was indicated that the solar radiation itself could potentially promote atmospheric mercury deposition on plants and may be related to stomatal conductance caused by the impacts.

During the reproductive stage of *L. chinensis* (Table 8), the direct effects of various environmental factors from high to low were as follows: Atmospheric humidity; solar radiation; soil humidity; and atmospheric mercury concentrations. The addition of other variables did not significantly improve the regression model. Among the aforementioned factors, the influential effects of atmospheric humidity on the mercury flux were the most dominant, which were strong negative effects. Among the other environmental factors, the direct influence coefficients of the solar radiation were also relatively high, which were strong positive effects.

The results of the path analysis during the reproductive stage were found to be in line with the previous conjecture. In other words, the appearance of changes in the plant/atmospheric mercury flux values during the month of August were dominated by humidity. The higher humidity levels weakened the mercury release and increased the mercury deposition. Moreover, it is worth noting that the conversion of solar radiation from a negative effect to a positive effect may also indicate that the mercury exchanges through the stomatal pathways weakened at that time, and the mercury release phenomenon caused by the reduction in HgII due to solar radiation was relatively increased. Furthermore, in regard to the influential effects of the stomatal and non-stomatal pathways on the mercury flux, there continues to be some controversy among researchers [63]. The results of this study revealed that the degrees of influence of those two pathways on the plant/atmospheric mercury flux values were not static. For example, during the vegetative reproduction stage, the stomatal pathways had more obvious effects, while during the plant reproduction stage, the non-stomata pathways had greater impacts. Those differences are related to the aging of the plant tissues and the increases in mercury concentration levels within the plants.

#### 4.3.2. Environmental Factors and Exchange Dynamics of the Mercury Flux in *Setaria*

In the vegetative reproduction stage of *S. viridis* (Table 9), the direct effects of various environmental factors from high to low were: atmospheric mercury concentration, atmospheric humidity, solar radiation, atmospheric temperature, and soil temperature. The addition of other variables did not significantly improve the regression model. Among them, the influence of the atmospheric mercury concentration on mercury flux had a dominant position, which was a strong negative effect. Among other environmental factors, the direct influence coefficient of atmospheric humidity and solar radiation was relatively high, which was a negative effect.

The correlation between *S. chinensis* and *L. chinensis* in the vegetative reproductive stage and environmental factors showed a high consistency, and both were dominated by atmospheric mercury concentration. Solar radiation and atmospheric humidity also showed negative effects. However, it can be seen from the direct coefficient of the atmospheric mercury concentration that the direct influence coefficient of the atmospheric mercury concentration on the mercury flux of *Setaria* was lower than the direct influence coefficient on the mercury flux of *L. chinensis*, which indicates that the sensitivity of *Setaria* to atmospheric mercury concentration may be low. Compared with *L. chinensis*, it is not easy to rapidly increase mercury deposition flux when the atmospheric mercury concentration increases.

In the reproduction stage of *S. viridis* (Table 10), the direct effects of various environmental factors from high to low were: atmospheric humidity, soil temperature, solar radiation, and the addition of other variables did not significantly improve the regression model. Among them, the influence of atmospheric humidity on the mercury flux was dominant, which was a strong negative effect. Among other influential factors, soil temperature had a strong negative effect, and solar radiation had a positive effect.

It can be seen that in the reproductive stage, the effect of humidity on the mercury flux of *S. chinensis* was consistent with the effect on the mercury flux of *L. chinensis*, and it showed a very strong negative effect, which was higher than that of any other influential factors on mercury in any period of time. This phenomenon shows that, in addition to mercury deposition, moisture is also an important limiting factor in the reproductive stage of *Setaria* in the study area. At the same time, it was found that the solar radiation and atmospheric humidity were not eliminated in the linear model fitting of the two plants during the two growth periods. This indicates that the solar radiation and atmospheric humidity had an effect on the mercury flux between the two plants and the atmosphere during the entire experimental period. The impacts were more important.

#### 4.3.3. Comparison of the Results of the Correlation Analysis between the two Plant Types and the Environmental Factors

From the results of the Pearson correlation analysis, it could be seen that the relationships between the environmental factors and the two plant types displayed the same trends. The results of the path analyses of the mercury flux for the two plant types during the vegetative and reproductive stages showed that the environmental factors that had the most effective influences were the solar radiation and atmospheric humidity. As can be seen in Table 7, Table 8, Table 9 and Table 10, the direct influence coefficients of the solar radiation on the mercury flux of *L. chinensis* were higher than those of *S. vulgaris* during the growth stages of the two types of plants. Meanwhile, the direct influence coefficients of the atmospheric humidity on the mercury flux of *L. chinensis* were higher than those of the sheep grass.

It was determined that solar radiation promotes atmospheric mercury deposition on plants during vegetative reproductive periods and promotes mercury release from plants to the atmosphere during the reproductive periods. It was also concluded that atmospheric humidity promotes atmospheric mercury deposition on plants during both growth stages. At the same time, the two types of plants had different effects on the atmospheric humidity. The sensitivity to humidity was observed to be higher than the sensitivity to solar radiation. Therefore, the influential effects of atmospheric humidity were considered to be more likely to change the source and sink abilities of the plants. The direct influence coefficients of the atmospheric humidity on the mercury flux of the two types of plants during the reproductive period were higher than the direct influence coefficients of the atmospheric humidity on the mercury flux of the two types of plants during the vegetative reproductive period. These findings indicated that atmospheric humidity promoted atmospheric mercury deposition on plants to a lesser extent during the months of June and July and inhibited mercury emissions from the plants to the atmosphere to a higher degree during the month of August. Therefore, the direct influence coefficients of the atmospheric humidity on *Setaria* in August were observed to be extremely high, higher than those of *L. chinensis*. As a result, it was indicated that during the mercury release stage of the plants, *Setaria* had a stronger dependence on the inhibitory effects of atmospheric humidity for the release of mercury. Howard et al. [39] pointed out that grassland vegetation is a source of atmospheric mercury when grassland plants age. Schonherr [26] also believed that as plant tissue ages, the absorption of atmospheric substances by the leaves will be reduced. Therefore, it can be considered that the vegetation of the *L. chinensis* grassland in this study changed from an atmospheric mercury sink to an atmospheric mercury source during the reproductive period and could potentially remain as an atmospheric mercury source until the end of the plant life cycle and the decomposition of the plant litter. Furthermore, since the annual precipitation in the experimental area is low and is concentrated during the months of June to September, the atmospheric humidity during the later period of the plant life history can be expected to decrease. This will cause the humidity levels to greatly reduce the inhibitory effects of the humidity on the mercury release processes of *S. viridis* and make *S. viridis* a source of atmospheric mercury. The role of the source will then be improved. In addition, due to the fact that the direct influence coefficients of the atmospheric humidity on *L. chinensis* were lower than those of *S. chinensis*, the influential effects of the humidity changes on *L. chinensis* were lower than those observed for *S. chinensis*.

Since the role of *S. chinensis* as a source of atmospheric mercury was greater in this study than that of *L. chinensis* during the mercury release stages, its role was considered to have a strong growth trend. Therefore, it can be expected that as the vegetation in the area degrades, the coverage of *Setaria* and other weed species will increase. As a result, the role of the grassland area as a sink of atmospheric mercury will be reduced.

## 5. Conclusions

In this study, *L. chinensis* and *S. serrata* were identified as the experimental plants. Plant and soil samples from the vegetative reproductive period to the reproductive period were collected over a three-month period. The characteristics of the plant and atmospheric mercury fluxes and related environmental factors were determined. The following conclusions were reached based on this study’s analysis results:1.During the mercury absorption periods, the mercury flux levels of *L. chinensis* and *S. chinensis* were observed to be basically the same. However, during the mercury release periods, the mercury flux levels of *S. chinensis* were found to be higher than those of *L. chinensis*.2.The day and night changes in the mercury flux values for the two plant types were high during the day and low at night, with the peaks occurring at approximately noon.3.*S. chinensis* was more inclined to participate in plant/atmospheric mercury exchanges than *L. chinensis.* It was speculated that during the community succession process in the grasslands, as the biomass ratio of *Setaria* and other weeds increases, the roles of the regional grasslands as atmospheric mercury sinks may weaken.4.The results of Pearson’s correlation and path analyses showed that the atmospheric mercury concentration levels were the main influential factors of the mercury fluxes during the vegetative reproductive periods of both examined plant types.5.The results of Pearson’s correlation and path analyses showed that the atmospheric humidity factor was the main influential factor during the reproductive periods of the two types of plants.

## Figures and Tables

**Figure 1 ijerph-18-10115-f001:**
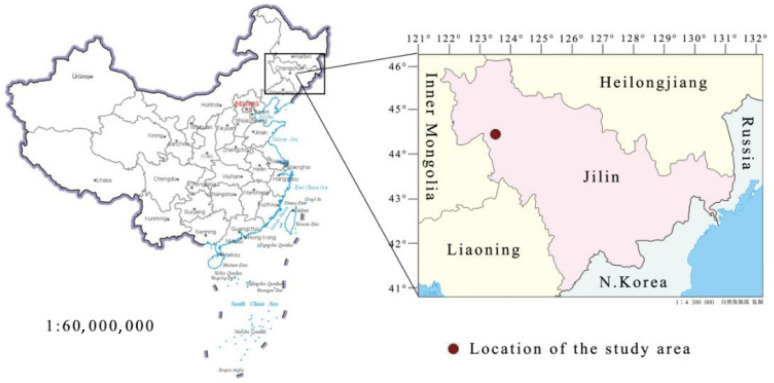
Location of the study area.

**Figure 2 ijerph-18-10115-f002:**
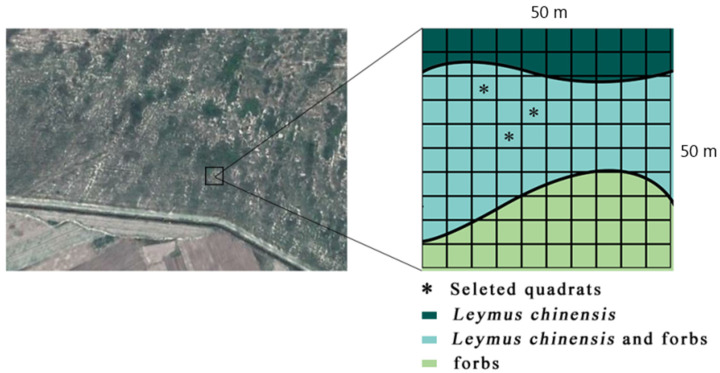
Sampling layout.

**Figure 3 ijerph-18-10115-f003:**
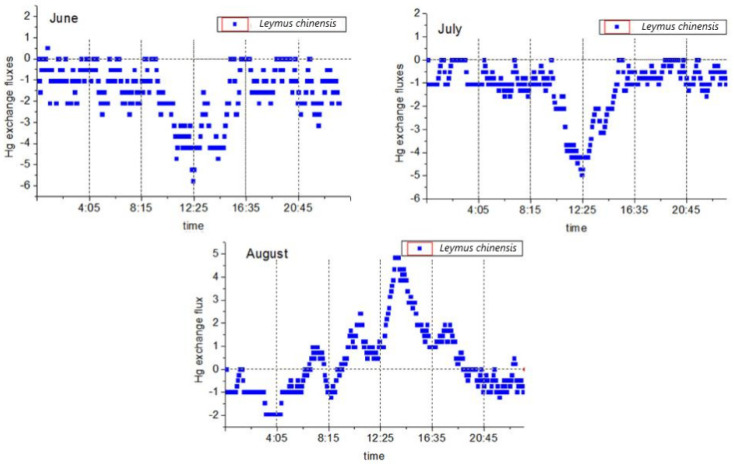
Changes in the daily mercury flux of *Leymus chinensis* during different months.

**Figure 4 ijerph-18-10115-f004:**
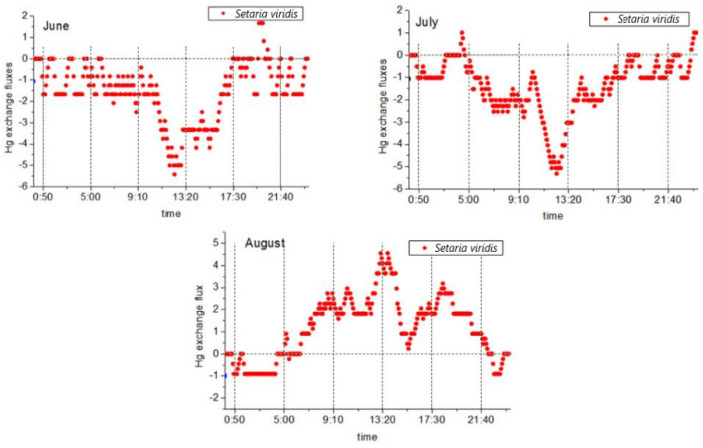
Changes in the daily mercury fluxes of *S. viridis* during different months.

**Table 1 ijerph-18-10115-t001:** Day and night concentrations of total gaseous mercury (ng·m^−3^).

Handle	Period	Scope	Daily Average
*L. chinensis*	6/26	4.0~28.0	16.6 ± 6.9
	7/21	7.0~28.0	16.6 ± 6.6
	8/22	8.0~28.0	16.7 ± 6.5
*S. sylvestris*	6/27	7.0~26.0	16.4 ± 6.0
	7/22	7.0~26.0	16.4 ± 6.0
	8/23	8.0~21.0	13.8 ± 4.2

**Table 2 ijerph-18-10115-t002:** Total soil mercury concentrations (ng·g^−1^).

Handle	Month		
	June	July	August
*L. chinensis* + weeds	16.3 ± 1.5	6.6 ± 0.7	8.9 ± 1.0

**Table 3 ijerph-18-10115-t003:** Independent processing results of the plant and atmosphere 24-h mercury exchange and deposition release fluxes (ng·m^−2^·h^−1^).

Species	Period	Mean Mercury Flux (Scope)	SD	Release Value	SD	n_1_	Sedimentation Value	SD	n_2_
*L. chinensis*	6/26	−1.58 (−5.76~0.52)	1.3	0.52	0.0	1	−1.59	1.3	288
	7/21	−1.19 (−4.97~0.00)	1.1	0	0.0	0	−1.19	1.1	289
	8/22	0.34 (−1.94~4.84)	1.5	1.63	1.4	134	−0.77	1.5	155
*S. vulgaris*	6/27	−1.54 (−5.42~1.67)	1.4	1.37	0.5	7	−1.65	1.3	282
	7/22	−1.37 (−5.31~1.01)	1.2	0.71	0.3	10	−1.46	1.2	279
	8/23	1.18 (−0.91~4.55)	1.4	1.99	1.0	193	−0.46	0.4	96

**Table 4 ijerph-18-10115-t004:** Plant indicators.

Species	Month	Biomass (g/m^2^)	Height (cm)	Leaf Area (m^2^)
*L. chinensis*	June	359.660	25.5	0.43
	July	412.356	26.8	0.86
	August	405.214	25.6	0.93
*S. vulgaris*	June	466.358	26.9	0.54
	July	529.936	37.6	0.89
	August	509.812	35.9	0.99

**Table 5 ijerph-18-10115-t005:** Relative growth rates.

Species	Time Period	
	June to July	July to August
*L. chinensis*	0.0055	−0.0005
*S. vulgaris*	0.0051	−0.0012

**Table 6 ijerph-18-10115-t006:** Analysis results of linear correlations between the impact factors and the mercury flux.

Impact Factor	Species	Growth Stage	r	*p*
Atmospheric mercury concentrations	*L. chinensis*	Vegetative reproduction	−0.517	0.000 **
		Reproduction	0.758	0.000 **
	*Setaria*	Vegetative reproduction	−0.598	0.000 **
		Reproduction	0.651	0.000 **
Temperature	*L. chinensis*	Vegetative reproduction	−0.523	0.000 **
		Reproduction	0.832	0.000 **
	*Setaria*	Vegetative reproduction	−0.579	0.000 **
		Reproduction	0.795	0.000 **
Humidity	*L. chinensis*	Vegetative reproduction	0.351	0.000 **
		Reproduction	−0.773	0.000 **
	*Setaria*	Vegetative reproduction	0.409	0.000 **
		Reproduction	−0.843	0.000 **
Sun radiation	*L. chinensis*	Vegetative reproduction	−0.559	0.000 **
		Reproduction	0.865	0.000 **
	*Setaria*	Vegetative reproduction	−0.670	0.000 **
		Reproduction	0.623	0.000 **
Soil temperature	*L. chinensis*	Vegetative reproduction	−0.391	0.000 **
		Reproduction	0.821	0.000 **
	*Setaria*	Vegetative reproduction	−0.504	0.000 **
		Reproduction	0.674	0.000 **
Soil humidity	*L. chinensis*	Vegetative reproduction	0.445	0.000 **
		Reproduction	−0.741	0.000 **
	*Setaria*	Vegetative reproduction	0.342	0.000 **
		Reproduction	−0.801	0.000 **

** *p* < 0.01.

**Table 7 ijerph-18-10115-t007:** Path analysis of the mercury flux and environmental factors during the vegetative reproduction stage of *Leymus chinensis*.

Environmental Factors	Direct Influence Coefficient	Indirect Influence Coefficient					
		Atmospheric Mercury Concentration	Humidity	Sun Radiation	Soil Temperature	Soil Humidity	Total
Atmospheric mercury concentration	−0.979	-	0.510	−0.396	0.503	−0.155	0.462
Humidity	−0.558	0.895	-	0.354	−0.497	0.158	0.91
Sun radiation	−0.496	−0.781	0.398	-	0.471	−0.152	−0.064
Soil temperature	0.564	−0.873	0.492	−0.414	-	−0.159	−0.954
Soil humidity	0.206	0.738	−0.428	0.366	−0.436	-	0.24

**Table 8 ijerph-18-10115-t008:** Path analysis of the mercury flux and environmental factors during the reproductive stage of *L. chinensis*.

Environmental Factors	Direct Influence Coefficient	Indirect Influence Coefficient				
		Sun Radiation	Humidity	Soil Humidity	Atmospheric Mercury Concentration	Total
Sun radiation	0.737	-	0.616374	−0.234	−0.253843	0.129
Humidity	−0.849	−0.535062	-	0.313625	0.297037	0.076
Soil humidity	0.325	−0.53064	−0.819285	-	0.283265	−1.067
Atmospheric mercury concentration	−0.313	0.597707	0.297037	0.283265	-	1.178

**Table 9 ijerph-18-10115-t009:** Path analysis of the mercury flux and environmental factors during the vegetative reproductive stage of *Setaria*.

Environmental Factors	Direct Influence Coefficient	Indirect Influence Coefficient					
		Atmospheric Mercury Concentration	Temperature	Humidity	Sun Radiation	Soil Temperature	Total
Atmospheric mercury concentration	−0.66	-	−0.27323	0.521853	−0.369005	0.182655	0.062
Temperature	−0.307	−0.5874	-	0.524808	−0.38493	0.17548	−0.272
Humidity	−0.591	0.58278	0.272616	-	0.32032	−0.175275	1
Sun radiation	−0.455	−0.53526	−0.259722	0.416064	-	0.164	−0.215
Soil temperature	0.205	−0.58806	−0.262792	0.505305	−0.364	-	−0.71

**Table 10 ijerph-18-10115-t010:** Path analysis of the mercury flux and environmental factors during the reproductive stages of *Setaria*.

Environmental Factors	Direct Influence Coefficient	Indirect Influence Coefficient			
		Humidity	Sun Radiation	Soil Temperature	Total
Humidity	−1.245	-	−0.286832	0.69732	0.41
Sun radiation	0.394	0.90636	-	−0.67626	0.23
Soil temperature	−0.78	1.11303	0.341598	-	1.45

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
