# Peer review of "The Characteristics of Mercury Flux at the Interfaces between Two Typical Plants and the Air in Leymus chinensis Grasslands"

_ijerph, 2021, doi:10.3390/ijerph181910115_

Round 1
Reviewer 1 Report
to put the interesting results into proper context, both the atmospheric lifetime and concentration of Hg in troposphere (2 ng/m3 ~ 230 ppqv [mixing ratio 2.3*10-13] in pure air) should be mentioned in the introduction already. Can the fact that humidity and Hg downward transport are strongly correlated be acribed to wet deposition involving hydrometeors? The correlation between upward mixing and heat is obvious given the volatility of Hg. Using the above values and evaporation enthalpy, mixing ratio of Hg, the standard potential for Hg(g)/Hg2+biomass of some 0.85 V vs. NHE is reduced to a value very close to that for CO2;HCO3-/CH4, explaining why methanogenesis and Hg venting are combined in e.g., bogs while many bacteria use Hg reduction to get rid of this poison.
I would like to read something in discussion which identifies possibly risky form of landuse, and an evaluation of the elevated (factor ~ 8 - 10) levels of Hg vapor at your test site in toxicological terms. To put it bluntly: is this a possible toxicological problem, or should Hg just be taken as a powerful tracer for biogeochemical processes?
Author Response
From: Gang Zhang
School of Environment, Northeast Normal University,
Key Laboratory of Vegetation Ecology, Ministry of Education, Northeast Normal University,
Institute of Grassland Science, Northeast Normal University,,
Changchun Jilin, 130117, China.
September 12 2021
Dear Reviewer:
Thank you for considering the revised version of our manuscript “The Characteristics of Mercury Flux at the Interfaces Between Two Typical Plants and the Air in Leymus chinensis Grasslands (Ms. ID.: ecology-1365062)”, for publication in Ecology. We appreciate all the comments from the reviewers on the previous manuscript. We have thoughtfully considered these comments. The following are the itemized responses to the reviewers’ comments (given in blue below). We highlighted all the changes by using a red font in the revised manuscript.
In addition to the reviewers’ revision suggestions, we also found some other shortcomings in the
manuscript and have corrected them in the revised manuscript. We also highlighted them by using a red font.
Thank you very much!
Yours sincerely,
Gang Zhang behalf of the authors.
Major points: (“C” means Comment, “R” means Response)
C1: Can the fact that humidity and Hg downward transport are strongly correlated be acribed to wet deposition involving hydrometeors? The correlation between upward mixing and heat is obvious given the volatility of Hg.
R1: Thanks very much for recommending us about intrinsic plant action from a micro perspective. Your thinking provides a new perspective for our further understanding and analyzing the role of grass plants in the grassland ecosystem. The flux between plants and the air is a two-way, when release is greater than absorption, the plant release to the atmosphere; when absorption is greater than release, the plant absorption is fixed from the atmosphere. Hg's saturated steam pressure of 20℃ of 1.201 × 10-3 mmHg, showed a similar global migration process to water. Hg, under solar radiation, is absorbed and fixed by the plant, and is reduced to the form of the zero valence state.
C2: Using the above values and evaporation enthalpy, mixing ratio of Hg, the standard potential for Hg(g)/Hg2+biomass of some 0.85 V vs. NHE is reduced to a value very close to that for CO2;HCO3-/CH4, explaining why methanogenesis and Hg venting are combined in e.g., bogs while many bacteria use Hg reduction to get rid of this poison.
R2: Thank you for asking questions and suggestions from the perspective of mercury methylation and demethylation, which will help us to understand this scientific issue. Mercury is fixed in the form of methylmercury and released as demethylated. We strongly agree that microbes have an important role in the process of demethylation. Plants do play a release role in some cases. We will later explain the detailed mechanism of action through other articles.
C3: To put it bluntly: is this a possible toxicological problem, or should Hg just be taken as a powerful tracer for biogeochemical processes?
R3: Thank you for asking questions and suggestions from the perspective of mercury methylation and demethylation, which will help us to understand this scientific issue. Mercury is fixed in the form of methylmercury and released as demethylated. We strongly agree that microbes have an important role in the process of demethylation. Plants do play a release role in some cases. We will later explain the detailed mechanism of action through other articles.

Reviewer 2 Report
Dear Editor/Authors,
The article prepared by the authors contains interesting research which providing data about the influence of factors, mainly atmospheric on emissions of mercury by plants in the grassland regions of the Songnen Plains in northeastern China. The problem of mercury presence in the environment is still significant, despite numerous activities aimed at reducing emissions. I suggest accepting this article for printing, however, some of its fragments should be more accurately explained. This applies especially to signatures under tables and figures. In Table 1 what means "date", in Table 2 – the 6, 7, 8 should be changed to June, July, August, or describe what it means in the signature under this table. In Table 6 - Please describe thoroughly how the correlations have been calculated, the level of the correlation coefficient and what the "stars" means.
In how many repetitions, measurements were made for individual markings.
Author Response
From: Gang Zhang
School of Environment, Northeast Normal University,
Key Laboratory of Vegetation Ecology, Ministry of Education, Northeast Normal University,
Institute of Grassland Science, Northeast Normal University,,
Changchun Jilin, 130117, China.
September 12 2021
Dear Reviewer:
Thank you for considering the revised version of our manuscript “The Characteristics of Mercury Flux at the Interfaces Between Two Typical Plants and the Air in Leymus chinensis Grasslands (Ms. ID.: ecology-1365062)”, for publication in Ecology. We appreciate all the comments from the reviewers on the previous manuscript. We have thoughtfully considered these comments. The following are the itemized responses to the reviewers’ comments (given in blue below). We highlighted all the changes by using a red font in the revised manuscript.
In addition to the reviewers’ revision suggestions, we also found some other shortcomings in the
manuscript and have corrected them in the revised manuscript. We also highlighted them by using a red font.
Thank you very much!
Yours sincerely,
Gang Zhang behalf of the authors.
Major points: (“C” means Comment, “R” means Response)
C1: Table 1: In Table 1 what means "date".
R1: We replaced: “The mean "date" with "period" to make the expression clearer.
C2: Table 2: The 6, 7, 8 should be changed to June, July, August, or describe what it means in the signature under this table.
R2: We changed 6, 7, 8 to June, July, August.
C3: Table 6: Please describe thoroughly how the correlations have been calculated, the level of the correlation coefficient and what the "stars" means.
R3: The correlations have been calculated by Pearson correlation coefficient; and the two "stars" means that significance level is 1%. The level of the correlation coefficient is between -1 and 1. 1 means the variable is completely positively correlated, 0 means irrelevant, and -1 means completely negatively correlated. Line 606-609.
C4: In how many repetitions, measurements were made for individual markings.
R4: The repetitions of measurements were made for individual markings present to Line 316-318 and Line 338-339.

Reviewer 3 Report
This study is focused on the mercury exchanges between vegetation and the atmosphere as a part of the global mercury cycle. It is well designed, raises some very interesting objectives for the knowledge of the mercury cycle and has been approached correctly.
The introduction is very interesting and correct, but a review of the studies carried out in other areas for grasslands is lacking. Although there are more studies for forests than for grasslands, there are studies of mercury fluxes in these types of ecosystems in the literature that could be included in the paper. Additionally, it would be advisable to use more recent bibliography.
The methodology used is adequate and provides satisfactory results. The discussion is well prepared and the conclusions respond to the objectives set by the authors.
It would be advisable to take care of the format of text, tables and figures. Some examples: dates are confused with numbers, it is more advisable to put month/day instead of month.day; be careful with the significant figures of the decimals (for example in Table 4 for biomass (g)); improve the use of semicolons and other forms of punctuation such as for example throughout lines 64-66; and others.
Any case, the article can be published with minor revision.
Author Response
From: Gang Zhang
School of Environment, Northeast Normal University,
Key Laboratory of Vegetation Ecology, Ministry of Education, Northeast Normal University,
Institute of Grassland Science, Northeast Normal University,,
Changchun Jilin, 130117, China.
September 12 2021
Dear Reviewer:
Thank you for considering the revised version of our manuscript “The Characteristics of Mercury Flux at the Interfaces Between Two Typical Plants and the Air in Leymus chinensis Grasslands (Ms. ID.: ecology-1365062)”, for publication in Ecology. We appreciate all the comments from the reviewers on the previous manuscript. We have thoughtfully considered these comments. The following are the itemized responses to the reviewers’ comments (given in blue below). We highlighted all the changes by using a red font in the revised manuscript.
In addition to the reviewers’ revision suggestions, we also found some other shortcomings in the
manuscript and have corrected them in the revised manuscript. We also highlighted them by using a red font.
Thank you very much!
Yours sincerely,
Gang Zhang behalf of the authors.
Major points: (“C” means Comment, “R” means Response)
C1: A review of the studies carried out in other areas for grasslands is lacking. Although there are more studies for forests than for grasslands, there are studies of mercury fluxes in these types of ecosystems in the literature that could be included in the paper. Additionally, it would be advisable to use more recent bibliography.
R1: Compared with forest ecosystem and grassland ecosystem, forest ecosystem can conserve water and play an obvious role in climate regulation and has huge biomass, so it has become a hot research topic today. At present, with little research on grassland ecosystems and more research on forest ecosystems, our goal of this study is to fill the gap here. In studies on mercury flux. Very few are measured using the methods in this paper. We also supplement the research literature on grassland ecosystems and mercury fluxes. Line285-286.
C2: It would be advisable to take care of the format of text, tables and figures. Some examples: dates are confused with numbers, it is more advisable to put month/day instead of month.day.
R2: We replaced: “month.day” with “month/day” to make the expression clearer.
C3: Table 4: Be careful with the significant figures of the decimals (for example in Table 4 for biomass (g)).
R3: We corrected the units of the biomass (g) and significant figures of the decimals for a more accurate description.
C4: Lines 64-66: Lmprove the use of semicolons and other forms of punctuation such as for example throughout lines 64-66; and others.
R4: We elevated the expression of Lines 64-66; and others.
